# LEARNING CROSS-LINGUAL SENTENCE REPRESENTATIONS VIA A MULTI-TASK DUAL-ENCODER MODEL

## ABSTRACT

A significant roadblock in multilingual neural language modeling is the lack of labeled non-English data. One potential method for overcoming this issue is learning cross-lingual text representations that can be used to transfer the performance from training on English tasks to non-English tasks, despite little to no task-specific non-English data. In this paper, we explore a natural setup for learning cross-lingual sentence representations: the dual-encoder. We provide a comprehensive evaluation of our cross-lingual representations on a number of monolingual, cross-lingual, and zero-shot/few-shot learning tasks, and also give an analysis of different learned cross-lingual embedding spaces.

## 1 INTRODUCTION

There has been a significant amount of recent work on developing models that can produce sentence representations that are useful for a number of language processing tasks (Kiros et al., 2015; Conneau et al., 2017; Subramanian et al., 2018; Logeswaran & Lee, 2018; Cer et al., 2018). However, these models are trained on largely monolingual data, and can thus only be used for tasks in a single language. A promising direction for extending the previous models to multiple languages is learning cross-lingual embedding spaces (Schwenk et al., 2017; Eriguchi et al., 2018; Singla et al., 2018), which could be used to transfer performance in one language to others.

We develop a novel approach for cross-lingual representation learning by combining the dual-encoder architectures used for learning sentence representations (Logeswaran & Lee, 2018; Cer et al., 2018) and for bi-text retrieval (Guo et al., 2018). By doing so, we learn representations that maintain state-of-the-art performance in tasks for a source language while *simultaneously* obtaining state-of-the-art performance in zero-shot learning tasks for a target language. For a given source-target language pair, we construct a multi-task training scheme using native source language tasks, native target language tasks, and a bridging source-target translation retrieval task to learn sentence representations that are aligned between the source and target languages. We then evaluate the learned representations on several monolingual and cross-lingual tasks, and also provide a graph-based analysis of the learned representations.

We find that multi-task training using additional monolingual tasks improves performance over models that only make use of parallel data on both cross-lingual semantic textual similarity (STS) (Cer et al., 2017) and Søgaard et al. (2018)'s cross-lingual eigen-similarity metric. The results show that the addition of monolingual data improves the embedding alignment of sentences and their translations. Furthermore, we find that cross-lingual training with additional monolingual data leads to far better transfer learning performance, and we show that our cross-lingual representations outperform state-of-the-art zero-shot learning models in sentiment classification and natural language inference.

## 2 MULTI-TASK DUAL-ENCODER MODEL

The core of our approach is the idea of modeling various tasks as ranking input-response pairs by encoding them via two encoders, with the crucial task for learning cross-lingual representations being *translation ranking*. For translation ranking, as well as for our other tasks, we take an input sentence $s_i^I$ and an associated response sentence $s_i^R$, and we seek to rank $s_i^R$ over all other possible response sentences $s_j^R \in \mathcal{S}^R$. To do so, we model the conditional probability $P(s_i^R \mid s_i^I)$ as:

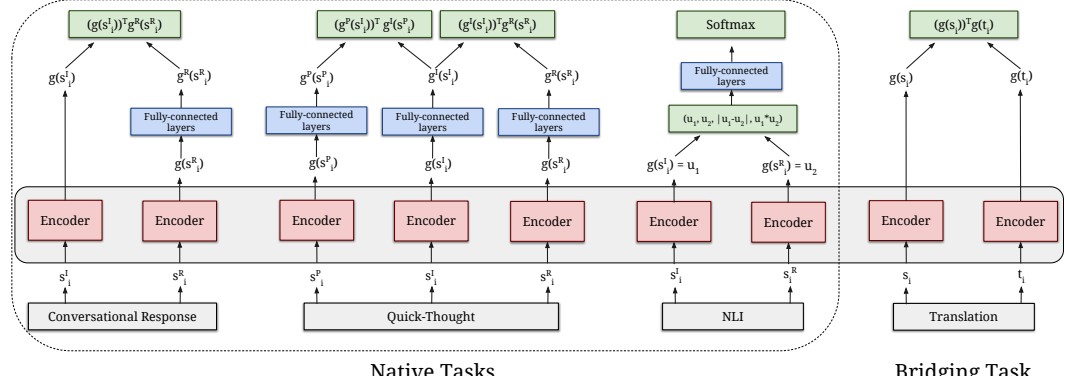

Figure 1: Multi-task dual-encoder model. It consists of a group of native tasks in each language and a bridging task using translation pair data. The encoders in the gray box all share their parameters, and thus constitute $g$.

$$P(s_i^R \mid s_i^I) = \frac{e^{\phi(s_i^I, s_i^R)}}{\sum_{s_j^R \in \mathcal{S}^R} e^{\phi(s_i^R, s_j^R)}}, \quad \phi(s_i^I, s_j^R) = g^I(s_i^I)^\top g^R(s_j^R) \tag{1}$$

Where $g^I$ and $g^R$ are the input and response sentence encoding functions that compose the dual-encoder. Since the normalization term in equation 1 is computationally intractable, we follow the approaches of Henderson et al. (2017) and instead choose to model an approximate conditional probability $\widetilde{P}(s_i^R \mid s_i^I)$:

$$\widetilde{P}(s_i^R \mid s_i^I) = \frac{e^{\phi(s_i^I, s_i^R)}}{\sum_{j=1, j \neq i}^{K} e^{\phi(s_i^R, s_j^R)}} \tag{2}$$

Where $K$ denotes the size of a single batch of training examples, and the $s_j^R$ correspond to the response sentences associated with the other input sentences in the same batch as $s_i^I$. We parametrize $g^I$ and $g^R$ as deep neural networks that are trained to minimize the negative log-likelihood of $\widetilde{P}(s_i^R \mid s_i^I)$ for each task.

In order to produce a single sentence encoding function $g$ that can be evaluated on downstream tasks, we share several layers between the input and response encoders and treat the final output of these shared layers as $g$. Additionally, these layers are modeled after the Universal Sentence Encoder (USE) model of Cer et al. (2018), since it is the state-of-the-art model that is most amenable to our setup. To learn cross-lingual representations, we train $g$ on several tasks from mirrored corpora across languages[1] for the source-target language pairs English-French (en-fr), English-Spanish (en-es), and English-German (en-de). The resulting model structure is illustrated in Figure 1.

## 2.1 ENCODER ARCHITECTURE

**Word and Character Embeddings.** As part of the training process for learning the cross-lingual sentence encoding function $g$, we learn embeddings for the words and characters present in the training data for a given source-target language pair. Word embeddings are learned end-to-end, as we noticed that pre-trained embeddings did not make a difference for final performance. Character embeddings are learned in a similar manner, but with the added stipulation that we consider character n-gram embeddings instead of single character embeddings by using a single feedforward layer with $tanh$ activation on top of character n-grams. Each word in an input sentence then obtains a character representation by having its character n-gram representations summed together. To have the sentence encoder $g$ leverage the word and character representations together without drastically increasing its number of parameters, we sum the word and character embeddings before using them as input to $g$.

---

[1]We explore some training setups without mirroring in the supplementary material.

**Transformer Encoder.** The actual architecture of the shared encoder $g$ consists of three[2] layers of transformer stacks, which contain the feed-forward and multi-head attention sub-layers described in Vaswani et al. (2017). The transformer encoder output is a variable-length sequence at each stack. We average encodings of all sequence positions in the final layer as the final sentence encoding. This embedding is then fed into different sets of feedforward layers that are used for each task. For our transformer layers, we use 8 attentions heads, a hidden size of 512, and a filter size of 2048.

## 2.2 MULTI-TASK TRAINING SETUP

To learn a function $g$ that is capable of strong cross-lingual matching and transfer learning performance for a source-target language pair while also maintaining monolingual downstream task performance, we employ four unique task types for each language pair. Specifically, we employ *a conversation response prediction task, a quick thought task, a natural language inference task*, and a bridging task – *translation ranking*. Six total tasks are used in training, as the first two tasks are mirrored across languages.

**Conversation Response Prediction.** We model the conversation response prediction task in the same manner as Yang et al. (2018). We minimize the negative log-likelihood of $\widetilde{P}(s_i^R \mid s_i^I)$, where $s_i^I$ is a single comment and $s_i^R$ is its associated response comment. For the response side, we model $g^R(s_i^R)$ as two fully-connected feedforward layers of size 320 and 512 with $tanh$ activation on top of $g(s_i^R)$. For the input side, however, we simply let $g^I(s_i^I) = g(s_i^I)$, as we noticed in early experiments that letting the optimization of the conversational response task more directly influence the parameters of the underlying sentence encoder $g$ led to better downstream task performance.

**Quick Thought.** We use a modified version of the Quick Thought task detailed by Logeswaran & Lee (2018). We minimize the sum of the negative log-likelihoods of $\widetilde{P}(s_i^R \mid s_i^I)$ and $\widetilde{P}(s_i^P \mid s_i^I)$, where $s_i^I$ is a sentence taken from an article and $s_i^P$ and $s_i^R$ are its predecessor and successor sentences respectively. For this task, we model all three of $g^P(s_i^P)$, $g^I(s_i^I)$, and $g^R(s_i^R)$ using separate, fully-connected feedforward layers of size 320 and 512 with $tanh$ activation on top of $g$, as we did for $g^R(s_i^R)$ in our conversational modeling task.

**Natural Language Inference (NLI).** We also include an *English-only* natural language inference task based on Bowman et al. (2015). For this task, we first encode an input sentence $s_i^I$ and its corresponding response hypothesis $s_i^R$ into vectors $u_1$ and $u_2$ using $g$. The vectors $u_1$, $u_2$ are then used to construct a feature vector $(u_1, u_2, |u_1 - u_2|, u_1 * u_2)$, where $(\cdot)$ represents concatenation and $*$ represents element-wise multiplication. The form of this feature vector is derived from the original experiments of Bowman et al. (2015). This feature vector is then fed into a single feedforward layer of size 512 that is used to perform the 3-way NLI classification.

**Translation Ranking.** Our translation task setup is identical to the one used by Guo et al. (2018) for bi-text retrieval. We minimize the negative log-likelihood of $\widetilde{P}(s_i \mid t_i)$, where $(s_i, t_i)$ is a source-target translation pair. Since the translation task is intended to align the sentence representations for the source and target languages, we do not use any kind of task-specific feedforward layers and instead use $g$ as both $g^I$ and $g^R$. Following Guo et al. (2018), we append 5 translations that are similar to the correct translation to each training example as "hard-negatives". Similarity is determined via a version of our model trained only on the translation ranking task. We did not see additional gains from using more than 5 hard-negatives.

## 3 EXPERIMENTS

### 3.1 CORPORA

We draw upon multiple, openly available data sources and training corpora for the training of the tasks mentioned above. For each of our datasets, we use 90% of the data for training, and the remaining 10% for development/validation. Our data preprocessing procedures are described in detail in the supplementary material.

---

[2] We tried up to six layers, but did not notice a significant difference beyond three.

**Reddit.** We preprocess the Reddit data extracted by Al-Rfou et al. (2016) into 600 million input-response comment pairs for training our conversation response prediction task. We also translate this data using the Google neural machine translation (NMT) system of Wu et al. (2016).

**Wikipedia.** To get native, non-English data, we extract triplets of contiguous sentences from English, French, Spanish, and German articles take from Wikipedia. Our final extracted corpus of Wikipedia sentence triplets consists of 127.9, 49.5, 29.8, and 49.3 million triplets for English, French, Spanish, and German respectively, which we use to train our Quick Thought task.

**Stanford Natural Language Inference (SNLI).** The NLI data we use is taken from the Stanford Natural Language Inference (SNLI) dataset of Bowman et al. (2015), which consists of 570K sentence pairs associated with one of three labels: *entailment*, *contradiction*, or *neutral*. The corpus is split into training (550K), validation (10K), and testing sets (10K).

**Translation.** The data for training the translation task is constructed using a system similar to the approach described by Guo et al. (2018). The final constructed corpus contains around 600M en-fr pairs, 470M en-es pairs and 500M en-de pairs.

## 3.2 MODEL CONFIGURATION

In all of our experiments, multi-task training is done by cycling through the different tasks (translation pairs, Reddit, Wikipedia, NLI) and performing an optimization step for a single task at a time. We train all of our models with a batch size of 100 using stochastic gradient descent with a learning rate of 0.008. All of our models are trained for 30 million steps. All input text is tokenized prior to being used for training. We build a vocab containing 200 thousand unigram tokens with 10 thousand hash buckets for out-of-vocabulary tokens. The character n-gram vocab contains 200 thousand hash buckets used for 3 and 4 grams. Both the word and character n-gram embedding sizes are 320. All hyperparameters are tuned based on performance on the development portion (random 10% slice) of our datasets. Finally, as an additional training heuristic, we multiply the gradients to the word and character embeddings by a factor of 100[3]. We found that using this embedding gradient multiplier alleviated vanishing gradient issues and greatly improved training.

We compare the proposed cross-lingual multi-task (referred to simply as multi-task) models with baseline models that are trained using only the translation ranking task, which we dub as the "translation-ranking" models.

## 3.3 MODEL PERFORMANCE ON ENGLISH DOWNSTREAM TASKS

We first evaluated all of our cross-lingual models on several downstream English tasks taken from SentEval (Conneau & Kiela, 2018) to verify the impact of cross-lingual training. Each task is described in the supplementary material. Results on the tasks are summarized in Table 1. We note that cross-lingual training does not hinder the effectiveness of our encoder on English tasks, as the multi-task models are close to state-of-the-art in each of the downstream tasks. For the Text REtrieval Conference (TREC) eval, we actually find that our multi-task models outperform the previous state-of-the-art models by a sizable amount.

## 3.4 CROSS-LINGUAL RETRIEVAL

We also evaluate both the multi-task and translation-ranking models' efficacy in performing cross-lingual retrieval by using held-out translation pair data. Following Henderson et al. (2017), we use precision at N (P@N) as the evaluation metric by checking if a source sentence's target translation ranks (where ranking is done using dot product) in the top $N$ scored candidates when considering $K$ other randomly selected target sentences. Unlike Henderson et al. (2017), we set $K$ to be 999 instead of 99 because using $K = 99$ results in all metrics quickly shooting up to 99%.

The translation-ranking model remains as a strong baseline for finding the true translation, with 95.4%, 87.5%, 97.5% P@1 for en-fr, en-es and en-de retrieval tasks respectively. The multi-task model performs almost identical with 95.1%, 88.8% and 97.8%, which provides empirical justifica-

---

[3]We tried different orders of magnitude for the multiplier and found 100 to work the best.

Table 1: Performance on classification transfer tasks.

| Model | MR | CR | SUBJ | MPQA | TREC | SST | STS Bench (dev / test) |
|---|---|---|---|---|---|---|---|
| *Cross-lingual Multi-task Models* | | | | | | | |
| en-fr | 77.9 | 82.9 | 95.5 | 89.3 | 95.3 | 84.0 | 0.803 / 0.763 |
| en-es | 80.1 | 85.9 | 94.6 | 86.5 | 96.2 | 85.2 | 0.809 / 0.770 |
| en-de | 78.8 | 84.0 | 95.9 | 87.6 | 96.1 | 85.0 | 0.802 / 0.764 |
| *Translation-ranking Models* | | | | | | | |
| en-fr | 68.7 | 79.3 | 87.0 | 81.8 | 89.4 | 74.2 | 0.668 / 0.558 |
| en-es | 67.7 | 75.7 | 83.5 | 86.0 | 94.4 | 72.6 | 0.669 / 0.631 |
| en-de | 67.8 | 75.2 | 84.4 | 83.6 | 86.8 | 74.6 | 0.673 / 0.632 |
| *State-of-the-art Models* | | | | | | | |
| InferSent | 81.1 | 86.3 | 92.4 | 90.2 | 88.2 | 84.6 | 0.801 / 0.758 |
| Skip-Thought LN | 79.4 | 83.1 | 93.7 | 89.3 | – | – | – |
| Quick-Thought | 82.4 | 86.0 | 94.8 | 90.2 | 92.4 | 87.6 | – |
| USE Transformer | 81.4 | 87.4 | 93.9 | 87.0 | 92.5 | 85.4 | 0.814 / 0.782 |

tion that it is possible to maintain embedding space alignment despite optimizing for native tasks in each individual language. We also experimented with P@3 and P@10, the results are identical.

## 3.5 MULTILINGUAL STS

We further test whether our learned cross-lingual representations can also perform well in their associated non-English language tasks by evaluating semantic textual similarity (STS) performance on French, Spanish, and German.

To evaluate Spanish-Spanish (es-es) STS, we use SemEval-2017 task 1 (STS17) track 3 of Cer et al. (2017), which contains 250 Spanish sentence pairs with human labeled similarity scores. We also evaluate es-en STS by using the track 4(a) task[4], which contains 250 en-es sentence pairs.

Beyond English and Spanish, however, there are no standard STS datasets available for other languages. As such, we evaluate on a translated version of the STS Benchmark dataset from Cer et al. (2017) for French, Spanish, and German. We use Google's translation system to translate the STS Benchmark sentences to French, Spanish and German. We believe that the results on our pseudo-multilingual STS Benchmark dataset are expected to still be a reasonable indicator of multilingual semantic similarly performance, since the NMT encoder-decoder architecture for translation differs significantly from our dual-encoder approach.

Following Cer et al. (2018), we first compute the sentence encodings $u, v$ of an STS sentence pair, and then score the sentence pair similarity based on the angular distance between the two vectors, $-\arccos\left(\frac{uv}{||u||\,||v||}\right)$. Table 2 shows the Pearson's correlation coefficient of the STS tasks for all models. The first column shows the trained model performance on original English STS Benchmark data. Columns 2 to 4 shows the the performance on the other languages. All multi-task models remain strong on the translated STS tasks, with around 0.77 for dev and 0.74 for test in all languages. Lastly, columns 5 and 6 shows the results of en-es models on STS17 tasks. The un-tuned multi-task models achieve 0.827 for the es-es task and 0.769 for the es-en task. As a point of reference, we also list the two best performing STS systems, Tian et al. (2017) (ECNU) and Wu et al. (2017) (BIT), reported from Cer et al. (2017). Our results are very close to these state-of-the-art feature engineered and mixed systems, which we describe in greater detail in the supplementary material.

## 4 ZERO-SHOT CLASSIFICATION

To evaluate the transfer learning capabilities of our models, we examine how well the multi-task and translation-ranking encoders perform on zero-shot and few-shot classification tasks.

---

[4]The es-en task is split into track 4(a) and track 4(b), we only use track 4(a) here. The track 4(b) task contains sentence pairs from WMT with only one annotator for each pair. The previous reported numbers are also particularly low for this task. We suspect it is not a very good evaluation set for current systems.

Table 2: Pearson's correlation coefficients on translated STS Benchmark and STS17 tasks. The first column shows the results on the original STS Benchmark data in English.

| Model | Translated STS Benchmark (dev / test) | | | | STS17 | |
|---|---|---|---|---|---|---|
| | en-en | fr-fr | es-es | de-de | es-es | es-en |
| Multi-task en-fr | 0.803 / 0.763 | 0.777 / 0.738 | – | – | – | – |
| Trans.-ranking en-fr | 0.668 / 0.558 | 0.641 / 0.579 | – | – | – | – |
| Multi-task en-es | 0.809 / 0.770 | – | 0.779 / 0.744 | – | 0.827 | 0.769 |
| Trans.-ranking en-es | 0.669 / 0.631 | – | 0.622 / 0.611 | – | 0.642 | 0.587 |
| Multi-task en-de | 0.802 / 0.764 | – | – | 0.768 / 0.722 | – | – |
| Trans.-ranking en-de | 0.673 / 0.632 | – | – | 0.630 / 0.526 | – | – |
| ECNU | – | – | – | – | 0.856 | 0.813 |
| BIT | – | – | – | – | 0.846 | 0.749 |

## 4.1 MULTILINGUAL NLI

We evaluate the zero-shot classification performance of our multi-task models on two multilingual natural language inference (NLI) tasks. However, prior to doing so, we first train a modified version[5] of our multi-task models that also includes training on the English Multi-genre NLI (MultiNLI) dataset of Williams et al. (2018) in addition to SNLI. We train with MultiNLI to be consistent with the baselines we compare to, which also use MultiNLI.

First, we make use of the professionally translated French and Spanish subsets of SNLI created by Agić & Schluter (2017) for an initial cross-lingual zero-shot evaluation of French and Spanish. We refer to these translated subsets as SNLI-X. There are 1000 examples in the translated subsets for each language. To evaluate, we simply feed the French and Spanish examples into the pre-trained English NLI sub-network of our multi-task models.

We also use the more recent dataset (XNLI) of Conneau et al. (2018), which provides a means for multilingual NLI evaluation in Spanish, French, German, Chinese and more. Since XNLI provides non-European-language evaluations, we also train an English-Chinese (en-zh) version of our multi-task model. There are 5000 examples in each XNLI test set, and zero-shot evaluation is once again done by feeding non-English examples into the pre-trained English NLI sub-network.

Table 3 lists the accuracy on the English SNLI test set as well as on SNLI-X and XNLI for all of our multi-task models. The original English SNLI accuracies are around 84% for all of our multi-task models, indicating that English SNLI performance remains stable in the multi-task training setting. The zero-shot accuracy on SNLI-X is around 74% for both the en-fr and en-es models. The zero-shot accuracy on XNLI is around 65% for en-es, en-fr, and en-de, and around 63% for en-zh, thereby significantly outperforming the pretrained sentence encoding baselines (X-CBOW) described in Conneau et al. (2018). The X-CBOW baselines use fixed sentence encoders that are the result of averaging tuned multilingual word embeddings.

Row 4 of Table 3 shows the zero-shot French NLI performance of Eriguchi et al. (2018), which is a state-of-the-art zero-shot NLI classifiers based on multilingual NMT embeddings. Our multi-task model shows comparable performance to the NMT-based model in both English and French.

## 4.2 AMAZON REVIEW

**Zero-shot Learning.** We also conduct a zero-shot evaluation based on the Amazon review data extracted by Prettenhofer & Stein (2010). We preprocess the Amazon reviews and convert the data into a sentiment classification task by considering reviews with strictly more than three stars as positive and strictly less than three stars as negative, in the same manner as Prettenhofer & Stein (2010). Each review contains a summary field and a text field, which we concatenate to produce a single input. As the multi-task models are trained with sentence lengths clipped to 64, we only take the first 64 tokens from the the concatenated text as the input. There are 6000 training reviews in English, which we split into 90% for training and 10% for development.

---

[5]Training with additional MultiNLI data did not significantly impact SNLI or downstream task performance.

Table 3: Zero-shot classification accuracy (%) on SNLI-X and XNLI datasets.

| Model | SNLI-X | | | XNLI | | | | |
|---|---|---|---|---|---|---|---|---|
| | en | fr | es | en | fr | es | de | zh |
| Multi-task en-fr | 84.2 | 74.0 | – | 71.6 | 64.4 | – | – | – |
| Multi-task en-es | 83.9 | – | 75.9 | 70.2 | – | 65.2 | – | – |
| Multi-task en-de | 84.1 | – | – | 71.5 | – | – | 65.0 | – |
| Multi-task en-zh | 83.7 | – | – | 69.2 | – | – | – | 62.8 |
| Eriguchi et al. (2018) (NMT en-fr) | 84.4 | 73.9 | – | | – | – | – | – |
| XNLI-CBOW zero-shot | – | – | – | 64.5 | 60.3 | 60.7 | 61.0 | 58.8 |
| *Non zero-shot baselines* | | | | | | | | |
| XNLI-BiLSTM-last | – | – | – | 71.0 | 65.2 | 67.8 | 66.6 | 63.7 |
| XNLI-BiLSTM-max | – | – | – | 73.7 | 67.7 | 68.7 | 67.7 | 65.8 |

We first encode inputs using the pre-trained multi-task and translation-ranking encoders and feed the encoded vectors into a 2-layer feed-forward network culminating in a softmax layer. We use layers of size 512 and $tanh$ activation functions in each layer. We use Adam for optimization with an initial learning rate of 0.0005 and a learning rate decay of 0.9 at every epoch during training. We use a batch size of 16 and train for 20 total epochs in all experiments. We freeze the cross-lingual encoder during training. The model architecture and parameters are tuned on the development set.

We first train the classifier on English data, and then evaluate it on the 6000 French and German Amazon review test examples. The results are summarized in Table 4. The accuracy on the English test set is 87.4% for the en-fr model and 87.1% for the en-de model, with the zero-shot accuracy being above 80% for both models. The translation-ranking models again perform worse on all metrics. Once again we compare the proposed model with Eriguchi et al. (2018), and find that our zero-shot performance has a reasonable gain on the fr test set[6].

Table 4: Zero-shot sentiment classification accuracy(%) on target language Amazon review test data after training on only English Amazon review data.

| Model | en | fr | de |
|---|---|---|---|
| Multi-task en-fr | 87.4 | 82.3 | – |
| Translation-ranking en-fr | 74.4 | 66.3 | – |
| Multi-task en-de | 87.1 | – | 81.0 |
| Translation-ranking en-de | 73.8 | – | 67.0 |
| Eriguchi et al. (2018) (NMT en-fr) | 83.2 | 81.3 | – |

**Few-shot Learning.** We further evaluate the proposed multi-task models via few-shot learning, by training on English reviews and only a portion of French and German reviews. Our few-shot models are compared with baselines of training on French and German reviews only. Table 5 shows the classification accuracy of the few-shot models, where the second row shows the percent of French and German data that is used when training each model. With as little as 20% of the French or German training data, the few-shot models perform nearly as good as the baseline models trained on 100% of the French and German data. Adding more French and German training data leads to further improvements in few-shot model performance, with the few-shot models reaching 85.8% accuracy in French and 84.5% accuracy in German when using all of the French and German data.

## 5    ANALYSIS OF CROSS-LINGUAL EMBEDDING SPACES

Motivated by the recent work of Søgaard et al. (2018) studying the graph structure of multilingual word representations, we perform a similar analysis for our learned cross-lingual sentence representations. To do so, we take $N$ samples of size $K$ from en-fr, en-es, and en-de translation data and then encode these samples using the corresponding multi-task and translation-ranking models. We then

---

[6]Eriguchi et al. (2018) also train a shallow classifier, but use only review text and truncate their inputs to 200 tokens. Our setup is slightly different, as our models can take a maximum of only 64 tokens.

Table 5: Sentiment classification accuracy(%) on target language Amazon review test data after training on English Amazon review data and a portion of French of German data. The second row shows the percent of French (fr) or German (de) data is used for training in each model.

| Model | fr | | | | | | de | | | | | |
|---|---|---|---|---|---|---|---|---|---|---|---|---|
| | 0% | 10% | 20% | 40% | 80% | 100% | 0% | 10% | 20% | 40% | 80% | 100% |
| Few-shot | 82.3 | 84.4 | 84.4 | 84.8 | 85.2 | 85.8 | 81.0 | 81.6 | 83.3 | 84.0 | 84.7 | 84.5 |
| Baseline | – | 79.2 | 80.0 | 82.7 | 84.3 | 84.9 | – | 75.5 | 77.7 | 81.6 | 83.5 | 84.4 |

compute pairwise distance matrices within each sampled set of encodings, and use these distance matrices to construct graph Laplacians[7]. Finally, we obtain the similarity $\Psi(S,T)$ between each model's source and target language embedding subsets by comparing the eigenvalues of the source language graph Laplacians to the eigenvalues of the target language graph Laplacians as follows:

$$\Psi(S,T) = \frac{1}{N} \sum_{i=1}^{N} \sum_{j=1}^{K} (\lambda_j(L_i^{(s)}) - \lambda_j(L_i^{(t)}))^2 \qquad (3)$$

Where $L_i^{(s)}$ and $L_i^{(t)}$ refer to the graph Laplacians of the source language and target language sentences obtained from the $i^{th}$ sample of source-target translation pairs. A smaller value of $\Psi(S,T)$ indicates higher eigen-similarity of the source language and target language embedding subsets. Following Søgaard et al. (2018) we use a sample size of $K = 10$ translation pairs, but we choose to use $N = 1000$ samples instead of $N = 10$ (as was done in their work) since we found $\Psi(S,T)$ to have very high variance at $N = 10$. The computed values of $\Psi(S,T)$ for our multi-task and translation-ranking models are summarized in Table 6.

Table 6: Average eigen-similarity values of source and target embedding subsets.

| Model | en-fr | en-es | en-de |
|---|---|---|---|
| multi-task | 0.592 | 0.526 | 0.761 |
| translation-ranking | 1.036 | 0.572 | 2.187 |

We find that the source and target embedding subsets constructed from the *multi-task models* exhibit greater average eigen-similarity than those resulting from the translation-ranking models for all source-target language pairs. This result is not necessarily intuitive, since one might expect the translation-ranking model to optimize more for alignment. Given that eigen-similarity correlates with the better performance of the multi-task models in almost all tasks, a potential direction for future work could be to introduce regularization penalties based on graph similarity in multi-task training. Interestingly, we also observe that the eigen-similarity gaps between the multi-task and translation-ranking models are not uniform across language pairs (although it may be that translation-ranking requires even more training). Thus, another direction could be to further study differences in the difficulty of aligning different source-target language embeddings.

## 6 CONCLUSION

In this work, we explored a straightforward framework for training cross-lingual, multi-task dual-encoder models. We showed that by training English-French, English-Spanish, and English-German multi-task models using our setup, we can achieve near-state-of-the-art or state-of-the-art performance in a variety of English tasks while also being able to produce similar caliber results in zero-shot transfer learning tasks for other languages. Finally, we note that the fact that multi-task training can actually improve performance on some downstream English tasks (TREC) is particularly interesting, and believe that there are many possibilities for future explorations of cross-lingual model training.

---

[7]See Zhang (2011) for an overview of graph Laplacians.

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

## A  Supplementary Material

### A.1  Data Preprocessing

In order to effectively use Reddit, Wikipedia, and translation data, we do a significant amount of preprocessing on the raw data. We describe our preprocessing procedures for each dataset in the following paragraphs.

**Reddit.** The raw Reddit corpus extracted by Al-Rfou et al. (2016) consists of 2.4 billion posts and comments from Reddit between 2007 and 2016, making it our largest data source by an order of magnitude. To preprocess this data for training our conversation response prediction task, we follow the same procedure as Yang et al. (2018). Essentially, we consider Reddit comments and their children (response comments) as input-response pairs. We filter out comments that have $\geq 350$ characters, due to the limitations on the number of input tokens our models can accept. We also remove comments that start with "https", "@", or "/r/", and also remove comments whose authors have "bot" in their usernames. Lastly, we find that Reddit comments also contain a small mix of non-English text, and we filter out comments where the percentage of alphabetic characters is $\leq 70\%$. As mentioned in the paper, the final, processed Reddit data consists of 600 million input-response pairs.

Since Reddit comments are predominantly English, we found the original data to be unsuitable for training encoders for non-English languages. To create conversational corpora for other languages, we translate the entire Reddit dataset using Google's neural machine translation (NMT) system of Wu et al. (2016).

**Wikipedia.** One concern with using the translated Reddit corpus as a monolingual task for non-English languages is that the translated data will propagate any errors made by the NMT system used for translation. To ameliorate this problem, we crawl Wikipedia and extract triplets of contiguous sentences from Wikipedia articles. We use Wikipedia due to it having well-formed articles with a broad coverage of several languages. We extract all Wikipedia articles for English, French, Spanish and German from the Wikipedia dump of May 5, 2018, and use the sentence splitter model of Gillick (2009) to split the articles into sentences. All article titles and article section names are treated as sentences inline. As mentioned in the paper, our final extracted Wikipedia corpus consists of 127.9 million English sentence triplets, 49.5 million French triplets, 29.8 million Spanish triplets, and 49.3 million German triplets for French, Spanish and German respectively.

**Stanford Natural Language Inference (SNLI).** We use the English Stanford Natural Language Inference (SNLI) dataset consisting of 570K sentence pairs of Bowman et al. (2015) as is, without any further preprocessing. We also use the provided splits for training (550K), validation (10K), and testing (10K).

**Translation.** The data for training the translation task is crawled from the public web using a system similar to the approach described by Uszkoreit et al. (2010). The extracted data is further cleaned by a pre-trained translation pair scoring system. We then generate "hard-negatives" following Guo et al. (2018) by using a pre-trained coarse translation-ranking model to determine translations that are close to a correct translation. As mentioned in the paper, the final constructed corpus contains around 600M en-fr pairs, 470M en-es pairs and 500M en-de pairs.

### A.2  Downstream Task Descriptions

Each of the English downstream tasks shown in Table 1 in the main body of the paper are described briefly below:

- **Movie Reviews (MR):** Multi-class prediction (scale of one to five stars) on movie review snippet data from Pang & Lee (2005).

- **Customer Reviews (CR):** Binary sentiment classification of sentences from customer reviews mined by Hu & Liu (2004).

- **Subjectivity (SUBJ):** Binary classification of sentences from movie reviews (Pang & Lee, 2004) as either objective or subjective.

- **Multi-Perspective Question Answering (MPQA):** Multi-class opinion polarity prediction on news data from Wiebe et al. (2005).
- **Text REtrieval Conference (TREC):** Multi-class classification of questions obtained from TREC by Li & Roth (2002).
- **Stanford Sentiment Treebank (SST):** Phrase-level binary sentiment classification of text from Socher et al. (2013).
- **Semantic Text Similarity (STS):** Semantic textual similarity of sentence pairs in the form of Pearson correlation with human judgments (Cer et al., 2017).

## A.3 STATE-OF-THE-ART STS SYSTEM DESCRIPTIONS

For the STS evaluations provided in the body of our paper, we compared to the state-of-the-art systems of Tian et al. (2017) and Wu et al. (2017). We refer the reader to Cer et al. (2017) for a detailed summary of the characteristics of these systems, but we provide a brief characterization below for convenience:

- **ECNU:** The ECNU model is an ensemble of random forest, gradient boosted decision trees, and deep learning models that leverages several manually constructed features such as n-gram overlap, edit distance, longest common prefix, common substrings, word alignments, and more.
- **BIT:** The BIT model relies mostly on a feature called information content (IC), which is based on the likelihood of occurrence of a concept (in this case, a word). Information content is then computed and combined for sentences through a hierarchical approach, which Wu et al. (2017) also augment with word embedding alignment to produce a final ensemble model.

## A.4 ABLATION TESTS AND FURTHER EXPERIMENTS

**No NLI.** Given that the multi-task models detailed in the body of the paper are trained with an English NLI task but no non-English NLI tasks, we evaluate how much training without this English NLI task affects model performance on downstream English tasks. The performance of these no-NLI models is summarized under the "Cross-lingual Multitask Transformer No SNLI" section of Table 7. We find that training without SNLI leads to comparable or better performance on all English downstream tasks except for STS, where we find training with SNLI provides a significant bump in performance.

**No Wikipedia.** We similarly investigate whether there is a significant advantage gained from using the only native target language data source, Wikipedia. To do so, we train en-fr, en-es, and en-de multi-task models without the Wikipedia-based Quick Thought task. The performance of these no-wiki models is shown under the "Cross-lingual Multitask Transformer No Wiki" section of Table 7. We find that training without the Wikipedia Quick Thought task does not have a strong impact on model performance, with some tasks being marginally better and others being marginally worse.

**No target language native tasks.** Additionally, the cross-lingual, multi-task models discussed in the body of our paper use a largely parallel task setup, where for each source-target language pair the monolingual target tasks are mirrored from the monolingual source tasks (the only exception being NLI, which is English-only). To test how useful this mirroring of monolingual tasks is, we train several models without target language monolingual tasks. We label these models as the "no non-English native task" models, and also summarize their downstream English task performance in Table 7. We note that removing the non-English native tasks actually leads to significant decreases in performance on TREC, SST, and STS. We find this quite interesting, as it provides some more empirical justification for the notion that cross-lingual training for a source-target language pair can actually improve monolingual source task performance.

Finally, we also evaluate the zero-shot performance of the no non-English native task models on Amazon review sentiment classification. As can be seen in Table 8, training without the non-English native tasks leads to lower performance on English and roughly the same zero-shot classification performance as training with only translation pair data.

Table 7: English task performance of different model configurations. Notably, removing non-English tasks in encoder training actually hurts performance on downstream English tasks.

| Model | MR | CR | SUBJ | MPQA | TREC | SST | STS Benchmark (dev / test) |
|---|---|---|---|---|---|---|---|
| *Cross-lingual Multitask Transformer* | | | | | | | |
| *SNLI + Native Tasks (report above)* | | | | | | | |
| en-fr | 77.9 | 82.9 | 95.5 | 89.3 | 95.3 | 84.0 | 0.803 / 0.763 |
| en-es | 80.1 | 85.9 | 94.6 | 86.5 | 96.2 | 85.2 | 0.809 / 0.770 |
| en-de | 78.8 | 84.0 | 95.9 | 87.6 | 96.1 | 85.0 | 0.802 / 0.764 |
| *Cross-lingual Multitask Transformer* | | | | | | | |
| *No SNLI* | | | | | | | |
| en-fr | 80.5 | 86.5 | 94.0 | 89.1 | 96.6 | 85.0 | 0.747 / 0.722 |
| en-es | 80.0 | 89.4 | 95.6 | 90.5 | 94.5 | 85.7 | 0.754 / 0.730 |
| en-de | 80.5 | 85.5 | 94.0 | 90.0 | 92.8 | 82.6 | 0.737 / 0.723 |
| *Cross-lingual Multitask Transformer* | | | | | | | |
| *No Wiki* | | | | | | | |
| en-fr | 78.9 | 85.8 | 92.1 | 90.0 | 94.7 | 82.7 | 0.804 / 0.774 |
| en-es | 79.1 | 87.4 | 94.5 | 89.6 | 94.3 | 83.0 | 0.809 / 0.769 |
| en-de | 79.6 | 85.9 | 94.1 | 89.7 | 92.2 | 82.5 | 0.801 / 0.741 |
| *Cross-lingual Multitask Transformer* | | | | | | | |
| *No non-English native tasks* | | | | | | | |
| en-fr | 79.4 | 84.0 | 93.5 | 89.7 | 93.8 | 83.4 | 0.797 / 0.758 |
| en-es | 77.4 | 81.7 | 94.9 | 89.4 | 94.0 | 82.3 | 0.796 / 0.761 |
| en-de | 79.3 | 83.7 | 94.0 | 88.1 | 91.2 | 82.5 | 0.760 / 0.732 |
| *Translation-ranking Models* | | | | | | | |
| en-fr | 68.7 | 79.3 | 87.0 | 81.8 | 89.4 | 74.2 | 0.668 / 0.558 |
| en-es | 67.7 | 75.7 | 83.5 | 86.0 | 94.4 | 72.6 | 0.669 / 0.631 |
| en-de | 67.8 | 75.2 | 84.4 | 83.6 | 86.8 | 74.6 | 0.673 / 0.632 |

Table 8: Zero-shot sentiment classification accuracy(%) on target language Amazon review test data after training on only English Amazon review data.

| Model | en | fr | de |
|---|---|---|---|
| Multi-task en-fr | 87.4 | 82.3 | – |
| Multi-task No non-English native tasks en-fr | 85.6 | 65.4 | – |
| Translation-ranking en-fr | 74.4 | 66.3 | – |
| Multi-task en-de | 87.1 | – | 81.0 |
| Multi-task No non-English native tasks en-fr | 85.1 | – | 67.2 |
| Translation-ranking en-de | 73.8 | – | 67.0 |

