# OpenReview forum: "Learning Cross-Lingual Sentence Representations via a Multi-task Dual-Encoder Model"
_ICLR.cc/2019/Conference_

### Official Review · AnonReviewer2 · 2018-10-30
**Limited novelty, strong evaluation, other languages and tasks?**

**Rating:** 6
**Confidence:** 5

**Review:**

The paper presents an intuitive architecture for learning cross-lingual sentence representations. I see weaknesses and strengths:

(i) The approach is not very novel. Using parallel data and similarity training (siamese, adversarial, etc.) to facilitate transfer has been done before; see [0] and references therein. Sharing encoder parameters across very different tasks is also pretty standard by now, going back to [1] or so.
(ii) The evaluation is strong, with a nice combination of standard benchmark evaluation, downstream evaluation, and analysis.
(iii) While the paper is on cross-lingual transfer, the authors only experiment with a small set of high-resource languages, where transfer is relatively easy.
(iv) I think the datasets used for evaluation are somewhat suboptimal, e.g.:
a) Cross-lingual retrieval and multi-lingual STS are very similar tasks. Other tasks using sentence representations and for which multilingual corpora are available, include discourse parsing, support identification for QA, extractive summarization, stance detection, etc.
b) Instead of relying on Agic and Schluter (2017), why don’t the authors use the XNLI corpus [2]?
c) Translating the English STS data using Google NMT to evaluate an architecture that looks a lot like Google NMT sounds a suspicious.
(v) While I found the experiment with eigen-similarity a nice contribution, there is a lot of alternatives: seeing whether there is a linear transformation from one language to another (using Procrustes, for example), seeing whether the sentence graphs can be aligned using GANs based only on JSD divergence, looking at the geometry of these representations, etc. Did you think about doing the same analysis on the representations learned without the translation task, but using target language training data for the tasks instead? The question would be whether there exists a linear transformation from the sentence graph learned for English while doing NLI, to the sentence graph learned for German while doing NLI.

Minor comments:
- “Table 3” on page 5 should be Table 2.
- Table 2 seems unnecessary. Since the results are not interesting on their own, but simply a premise in the motivating argument, I would present these results in-text.

[0] http://aclweb.org/anthology/W18-3023

---

> ### Author Response · Authors · 2018-11-09
> **Addressing comments and clarifications**
>
> Thank you for your comments and thoughtful questions. We address each comment individually below:
>
> Addressing Major Comments
> (I) Novelty. While we agree that we have not introduced a new architectural component in our cross-lingual multi-task models, we believe that our combination of current SOTA language modeling components and the accompanying analysis still raises interesting questions and demonstrates strong enough results to motivate new research.
>
> (II) Evaluations. Thank you - we also plan to add further evaluations (i.e. comparing to more lightweight encoder architectures such as the Deep Averaging Network of Iyyer et al. (2015)) to the supplementary material in the next revision of our paper.
>
> (III) Language Pairs. Our main reason for choosing English-Spanish, English-French, and English-German language pairs was the fact that these language pairs had a number of pre-existing evaluations available.
>
> (IV) Datasets.
> (a) We are in the process of experimenting with further evaluations (paraphrases, summaries) in more languages, and hope to add these evaluations to future revisions of our paper.
> (b) XNLI was not available at the time we were preparing the initial draft of this paper, but we plan to include evaluations on XNLI in our next revision. We do have some preliminary XNLI results using our trained English-French model, which shows an accuracy of 69% on English and 64.5% on French.
> (c) We agree that translated STS is not intended to be a surefire evaluation of STS performance in target languages, but evaluating sentence representations on translated datasets has been considered before (Eriguchi et al., 2018). Additionally, the Google NMT architecture uses an encoder-decoder structure as opposed to our dual-encoder architecture, so we felt there were sufficient enough differences between our approaches that evaluating on translated data may still provide some insight into STS performance in non-English languages.
>
> (V) Embedding Space Analysis. We absolutely agree that there are a number of interesting and different analyses that can be done on the learned sentence embedding spaces; our reason for using the eigen-similarity analysis was to extend the previous work done for word embeddings. We did consider other approaches, such as aligning the sentence embeddings, but we ultimately felt that a proper treatment of the many different analyses techniques that are possible for the embedding spaces would be outside of the scope of this work. We found your suggestion about performing the eigen-similarity analysis without translation data and only the monolingual tasks to be interesting, and an oversight on our part for not including in the initial version of our paper. We plan to include this evaluation in the next revision of our paper.
>
> Addressing Minor Comments
> *All table numbers should now read correctly.
>
> *We have greatly simplified Table 2 as suggested.

---

### Official Review · AnonReviewer1 · 2018-11-02
**It is an interesting paper which explores multi-task model for simultaneously improving both monolingual and cross-lingual tasks. However, due to missing information and lacking of clarity makes it hard to accept at this point of time.**

**Rating:** 4
**Confidence:** 4

**Review:**

Summary
----------
In this paper, authors explore learning of cross-lingual sentence representations with their proposed dual-encoder model. Evaluation conducted with learned cross-lingual representations on several tasks such as monolingual, cross-lingual, and zero-shot/few-shot learning show the effectiveness of the proposed approach. Also, they show provide a graph-based analysis of the learned representations.

Three positive and negative points of the paper is presented as follows:

pros
------

1. cross-lingual representation learning by combinining ideas from learning sentence representations and cross-language retrieval.
2. Multi-task setup of different tasks for improving cross-language and monolingual tasks.
3. Lot of experimental results.


cons
-----
1. Claim it works for monolignual tasks in target language such as zero-shot learning for sentiment classification and NLI. Also, for cross-lingual STS and eigen-similarity metric is hard to retrieve from the paper.

2. Many terms, datasets are used without being referenced.

3. Usage of existing approaches to build a single model for many tasks.

comments to authors
-----------------------


1. Dual-encoder architecture is inspired from Guo et al (2018) which uses encoding of source and target sentence with Deep neural network. However, it is here replaced into multi-task dual-encoder model.

2. What are the tasks that are very specific to source language?

3. Equation-1 is basically a logistic regression or softmax over \phi. However \phi is dot product of encodings as similar to Deep averaging networks (Iyyer et al. 2015) ?

4. In Section-2, it is unclear what does symmetric tasks mean? They use parallel corpora?

5. In Section-2.1, it is mentioned that Word embeddings are learned end-to-end. Does this mean they are not initialized with pretrained ones?

6. In Section-2.1, it is mentioned that word and character embeddings are learned in a computationally efficient way, what does it represent? They use less parameters, parallelizable?

7. Why only three layers of transformer, It is understood that 6-12 layers is required for effective encoding of sentences (Al-Rfou et al., 2018)

8. In model configuration, how is convergence decided. Any stopping criterion?

9. What are the splits for reddit, wikipedia datasets?

10. In Table-1, what does MR,CR etc., refer to? They are never mentioned before. Does all tasks only use only English ?



Overall it is an interesting paper which explores multi-task model for simultaneously improving both monolingual and cross-lingual tasks. However, due to missing information and lacking clarity in some details it is hard to accept at this point of time.

Minor issues
--------------

1. Sentences are very long and not easily comprehensible.
2. Target language and repsonse are used without referencing each other. Better to use one of them for better tracking.
3. No common notation for the model. It is been referenced with different names (cross-lingual multi-task model, multi-task dual-encoder model).

---

> ### Author Response · Authors · 2018-11-09
> **Clarifying datasets, language, and claims**
>
> Thank you for your many comments and suggestions - we have addressed each individual comment below:
>
> Addressing Major Cons
> 1. Monolingual/Cross-lingual Claims. Our intention with including the zero-shot SNLI and Amazon sentiment classification tasks was to show that a cross-lingual model could leverage English data to perform well on target language tasks even with very little target language data for the task. We did not mean to claim that our cross-lingual models would perform better than optimized, monolingual target language models; we apologize for not being clearer about this.
>
> 2. Dataset References. Thank you for bringing this oversight to our attention, we have updated the discussion of table 1 with a reference to SentEval and a pointer to the supplementary material, which has brief descriptions of the different tasks along with their accompanying references.
>
> 3. Usage of Existing Approaches. The purpose of our work in this paper was to investigate how different ideas from SOTA models could be combined in a way that could lead to effective cross-lingual representations that are useful in many scenarios. While we agree that we do not introduce a new novel architectural component ourselves, we believe that our experiments and accompanying analysis are sufficiently interesting for motivating new research in cross-lingual and multi-lingual sentence representations (i.e. eigen-similarity-based regularization).
>
> Addressing Comments
> 1. Our encoding architecture is inspired by the work of Guo et al. (2018) as well as Cer et al. (2018) and Logeswaran & Lee (2018) (all cited in introduction). We consider our proposed training setup to build upon the previous work by introducing multiple tasks across languages that are connected via translation.
>
> 2. As mentioned in section 2, the only task that is specific to the source language (English) is SNLI, and we have made this more clear in the revision.
>
> 3. While the DAN model of Iyyer et al. (2015) also uses a softmax, the DAN paper does not mention the dual-encoder-style response ranking approach with dot products to the best of our understanding.
>
> 4. Yes, symmetric tasks was intended to mean that we used the same corpora across languages (i.e. Wikipedia for source and target languages). We have reworded that line in the revision to make it clearer.
>
> 5. This is correct, we do not initialize using pre-trained word embeddings and instead learn all embeddings from scratch. We noticed that using pre-trained embeddings made practically no difference in final performance, which we now make clear in the section discussion word and character embeddings.
>
> 6. In bringing up computational efficiency in our discussion of word and character n-gram embeddings, we meant to draw attention to the fact that we pool embeddings. Choosing to not pool embeddings would have greatly increased the parameter count of our model.
>
> 7. The primary reason we used a relatively shallow transformer model was to keep the number of parameters in our cross-lingual, multi-task model in the same ballpark as other SOTA monolingual models. Additionally, we did not notice a significant gain from training with 4, 5, or 6 transformer layers instead of 3. We note that the Universal Sentence Encoder model shown in table 1 uses 6 transformer layers, and our cross-lingual models have comparable performance to it. That being said, we are also trying to add results for much deeper transformer networks (as in Al-Rfou et al., 2018) in the next revision of our paper.
>
> 8. Our apologies for not making the notion of convergence more clear; convergence is determined to be 30M steps for all cases (as that was also when training stabilized). This has been cleared up in the revision.
>
> 9. We also apologize for not making all of our dataset splits more clear; we have updated the paper with this information.
>
> 10. The lack of dataset detail was a large oversight on our part - we have made all of our monolingual English evaluations clear and added the necessary citations in our revision.
>
> Addressing Minor Issues
> 1. We have gone back over the paper and tried to simplify/remove sentences when it is possible to do so.
>
> 2. We have made sure the use of “target language” is consistent throughout the paper, so that there is no confusion between terms.
>
> 3. Similarly, we have also tried to clear up the names used for referencing our different model types so that they are as consistent as possible.

---

### Official Review · AnonReviewer3 · 2018-11-06
**A new framework for cross-lingual sentence representation which is an interesting mix of standard building blocks, but more convincing experiments are needed to appreciate the main contributions.**

**Rating:** 7
**Confidence:** 4

**Review:**

This paper proposes a novel cross-lingual multi-tasking framework based on a dual-encoder model that can learn cross-lingual sentence representations which are useful in monolingual tasks and cross-lingual tasks for both languages involved in the training, as observed on the experiments for three language pairs. The main idea of the approach is to model all tasks as input-response ranking tasks and introduce cross-lingual representation tying through the translation ranking task, introduced by Guo et al. (2018). All components of the framework are quite standard and deja-vu, but I like the paper in general, and the results seem quite encouraging. I have several comments on how to further strengthen the paper and improve the presentation of the main findings.

The proposed framework does not offer any substantial modeling contribution (i.e., all major components are based on SOTA models), but the framework is still quite interesting as a mixture of these SOTA components. I believe that some additional experiments would make the main contributions clearer and would also provide additional insights into the main properties of the proposed framework: 1) cross-linguality and 2) multi-tasking.

*Most of all, I am surprised not to see any ablation studies. For instance, what happens if we remove one of the two monolingual tasks in each language? How does that reduced model compare to the full model? Which monolingual task is more beneficial for the final performance in downstream tasks? Can we think of adding another monolingual task to boost performance further? I think that this sort of experiment would be more beneficial for the paper than a pretty long analysis from Section 5 (this analysis is still valid, but should be shortened substantially). Evaluating only multi-tasking without any cross-lingual training would also be very beneficial to recognise the extent of improvement achieved by adding cross-linguality to the model.

*How much does the proposed architecture depend on the choice of the encoding model for the function g? Have the authors experimented with other (recent and (near-)SOTA) encoding models? I would like to see a comparative analysis of this 'hyper-parameter'.

*I would like to see more experiments on more distant language pairs. This would make the paper even more interesting imho. I am also curious whether there would be a drop in performance reported conditioned on the distance/proximity between two languages in a language pair.

*I would like to see a more detailed description of the two best performing STS systems (ECNU and BIT). In what respect are these systems state-of-the-art feature engineered and mixed? I am not sure what this means without providing any additional context to the claim and description.

*How does the monolingual English STS model trained with the cross-lingual multi-task framework compare to the work of Conneau et al. (EMNLP 2017) which also used SNLI as the task on which to learn universal sentence representations. This would be a good experiment imho as it would show how much we gain from cross-lingual training and multi-tasking.

Minor:
*Page 3: Could you add a short footnote discussing how hard-negatives for the translation ranking task are selected? How do you compute similarity here?
*Do you expect performance to improve further by training MultiNLI instead of SNLI (or combining the two datasets)?
*"All hyperparameters are tuned based on preliminary experiments on a development set." -> What is used as the development set? More details needed.
*"Finally, as an additional training heuristic, we multiply the gradients to the word and character embeddings by a factor of 100." -> How is the value for the embedding gradient multiplier determined? Is there an automatic procedure to fine-tune this hyper-parameter or has this been done in a completely empirical way?
*Table 1: please define the task abbreviations before showing them in the table. It is not clear what each task is by relying only on the abbreviation.
*This dataset was not available at the time of the submission, but for the revision it would make sense to also evaluate on the new XNLI dataset of Conneau et al. (EMNLP 2018) for multilingual NLI experiments.

(After the first revision) I have raised the score after the very detailed author response (thanks for that!), but this is also conditioned on the authors making the actual revisions promised in their response. I am still quite interested to check how well the method works in a setup with distant language pairs.

---

> ### Author Response · Authors · 2018-11-09
> **Addressing clarifications and currently working on new evaluations**
>
> Thank you for taking the time to write such detailed feedback about our paper. We address each of your concerns below:
>
> Addressing Main Concerns
> *Ablation Studies. We think this concern is spot on, and plan to add a much more detailed analysis of how each monolingual task contributes to our cross-lingual multi-task setup to the next revision of our paper (we are trying these experiments now). Currently, we do have experiments where we remove mirrored corpora in the supplementary material of our paper. When we wrote the initial version of our paper, we chose to prioritize many different evaluations of our cross-lingual models over more analysis of the training setup due to time and page constraints, and we acknowledge that this was an oversight on our part.
>
> *Encoder Architecture. Our choice of the transformer architecture for our experiments was based on many recent SOTA results in different language modeling tasks coming from the use of transformer architectures (Al Rfou et al., 2018; Cer et al., 2018), but we agree that it is possible for other architectures to potentially perform better in our setup. We did run some initial experiments with LSTMs and Deep Averaging Networks (Iyyer et al., 2015) and found that they did not outperform transformer models. Additionally, the main inhibiting factor in using different encoding architectures was the multiplicative effect that it would have had on our model evaluations/analysis, which we felt would have been infeasible for conference submission.
>
> *Distant Language Pairs. We have also been interested in extending our experiments to more language pairs, and plan to add experiments with English/Non-European language pairs to the next revision of our paper as well. We focused on English-French, English-Spanish, and English-German language pairs in our first draft due to having the highest number of standard evaluations available for these language pairs.
>
> *SOTA STS Systems. We have updated the paper to better explain how ECNU and BIT work (details are now in the supplementary material), which we hope clarifies the complexity of these systems relative to our own.
>
> *Comparison with InferSent. The comparison between our cross-lingual models and InferSent (Conneau et al., 2017) is actually done in Table 1, and we apologize for not making that more clear in our discussion. Adding cross-lingual multi-task training led to better performance in TREC, SUBJ, and SST, and worse performance in MR, CR, and MPQA.
>
> Addressing Minor Concerns
> *Our apologies for not making the hard negative similarity computation more clear - we had some additional information about it in the supplementary material, and we have now moved this info to the main body of the paper.
>
> *Given the performance of the no-SNLI models in the supplementary material, we did not expect MultiNLI data to make a significant impact on cross-lingual model performance on downstream tasks.
>
> *We apologize for not clarifying how our data splits were done, that was another oversight on our part. As a development set, we took a 10% slice of each of our datasets - this information has now been made clear in the body of the paper.
>
> *We have clarified all of the task abbreviations and added the relevant citations for each task (all in supplementary material), to make table 1 more understandable.
>
> *In the next revision of our paper, we plan to include performance on XNLI (once all of these experiments have finished running). We do have some initial experiment using the trained English-French model and fine-tuning using MultiNLI data with frozen encoders. Our initial XNLI evaluation shows an accuracy of 69% on English and 64.5% on French.

---

### Author Response · Authors · 2018-11-09
**Initial revision with clarifications**

Thanks to all of the reviewers for their helpful comments. We have updated our paper with an initial revision that handles the following main requests:

* Simplifies tables and language when possible.
* Further clarifies datasets and training procedures.
* Adds more detail concerning the systems being compared to.

We will be looking to make future revisions that also include more evaluations, particularly focusing on adding XNLI with evaluations on English/Non-European language pairs.

---

### Author Response · Authors · 2018-11-26
**Secondary revision with XNLI evaluation and ablation tests**

Hi all, we have updated our paper to include evaluations on XNLI in the main body, as per the suggestions of the reviewers. We have also included further ablation tests in the supplementary material. Thank you all again for your comments!

---

### Meta-Review · Area_Chair1 · 2018-12-14
**interesting but not very novel framework**

**Confidence:** 4
**Recommendation:** Reject

**Metareview:**

Pros:

- A new framework for learning sentence representations
- Solid experiments and analyses
- En-Zh / XNLI dataset was added, addressing the comment that no distant languages were considered; also ablation tests

Cons:

-  The considered components are not novel, and their combination is straightforward
-  The set of downstream tasks is not very diverse (See R2)
-  Only high resource languages are considered (interesting to see it applied to real low resource languages)

All reviewers agree that there is no modeling contribution.  Overall, it is a solid paper but I do not believe that the contribution is sufficient.